# Patient-Specific 3D-Print Extracranial Vascular Simulators and Infrared Imaging Platform for Diagnostic Cerebral Angiography Training

**DOI:** 10.3390/healthcare10112277

**Published:** 2022-11-14

**Authors:** Te-Chang Wu, Jui-Yu Weng, Chien-Jen Lin, Yu-Kun Tsui, Jinn-Rung Kuo, Pei-Jarn Chen, Jhi-Joung Wang

**Affiliations:** 1Department of Radiology, Chi-Mei Medical Center, Tainan 71004, Taiwan; 2Department of Medical Sciences Industry, Chang Jung Christian University, Tainan 71101, Taiwan; 3Department of Electrical Engineering, Southern Taiwan University of Science and Technology, Tainan 71005, Taiwan; 4Department of Medical Research, Chi-Mei Medical Center, Tainan 71004, Taiwan; 5Department of Neurosurgery, Chi-Mei Medical Center, Tainan 71004, Taiwan; 6Allied AI Biomed Center, Southern Taiwan University of Science and Technology, Tainan 71004, Taiwan

**Keywords:** 3D printing, infrared imaging, cerebral angiography, simulator, neuro-interventionalist

## Abstract

Tortuous aortic arch is always challenging for beginner neuro-interventionalists. Herein, we share our experience of using 3D-printed extracranial vascular simulators (VSs) and the infrared imaging platform (IRIP) in two training courses for diagnostic cerebral angiography in the past 4 years. A total of four full-scale patient-specific carotid-aortic-iliac models were fabricated, including one type I arch, one bovine variant, and two type III arches. With an angiography machine (AM) as the imaging platform for the practice and final test, the first course was held in March 2018 had 10 participants, including three first-year residents (R1), three second-year residents (R2), and four third-year residents (R3). With introduction of the IRIP as the imaging platform for practice, the second course in March 2022 had nine participants, including 3 R1s, 3 R2s, and 3 R3s. The total manipulation time (TMT) to complete type III aortic arch navigation was recorded. In the first course, the average TMT of the first trial was 13.1 min. Among 3 R1s and 3 R2s attending the second trial, the average TMT of the second trial was 3.4 min less than that of the first trial. In the second course using IRIP, the average TMT of the first and second trials was 6.7 min and 4.8 min, respectively. The TMT of the second trial (range 2.2~14.4 min; median 5.9 min) was significantly shorter than that of the first trial (range 3.6~18 min; median 8.7 min), regardless of whether AM or IRIP was used (*p* = 0.001). Compared with first trial, the TMT of the second trial was reduced by an average of 3.7 min for 6 R1s, which was significantly greater than the 1.7 min of R2 and R3 (*p* = 0.049). Patient-specific VSs with radiation-free IRIP could be a useful training platform for junior residents with little experience in neuroangiography.

## 1. Introduction

In the past two decades, neurovascular intervention has evolved rapidly and constituted the mainstream treatment of various neurovascular diseases, including procedures such as endovascular embolization for hemorrhagic neurovascular lesions (cerebral aneurysms, arteriovenous malformation, and dural arteriovenous fistula) [1], stenting for carotid stenosis [2] and intracranial atherosclerosis disease [3], and endovascular thrombectomy of acute ischemic stroke [4]. In the real world, however, there is little training equipment to help junior trainees become familiar with interventional neurovascular techniques and devices [5]. Traditionally, neurovascular residency training is based on the clinical narratives and the procedure observation in the angio-suite. It is even tougher for the clinicians without the experience of catheter manipulation. Building up a stable proximal access to the ipsilateral carotid artery is the first and mandatory step for the success of all neuroendovascular procedures. However, the tortuous type III aortic arch (Figure 1) and bovine aortic variant are always challenging for beginner neuro-interventionalists [6,7]. This can be attributed to a lack of experience in managing these strenuous arch anatomies with relatively low prevalence in daily practice. There are few vascular simulators (VSs) for treatment planning and little repetitive practice for achieving an uneventful procedure. 

With the advancement of 3D-print techniques, the medical simulation model is becoming an important modality in neurosurgical training. In the field of neurovascular diseases, 3D-print medical simulators have been applied for the informed consent of neurovascular surgery [8], the teaching course for skull base anatomy and skull base surgery simulation [9], and the preoperative simulation of aneurysm microsurgery [10,11,12,13,14]. Recently, a patient-specific neurovascular simulator with hollow channels has been fabricated for the training [15], preoperative simulation [16,17,18,19,20] of neurovascular intervention, and fluid dynamic analysis [21,22,23]. As compared with the traditional training methods and animal models, patient-specific vascular simulators can provide a tailored platform for repetitive practice with different vascular anatomies and allow for familiarity with the manipulation of microcatheters/microwires. Total manipulation time has been proposed as a measure of the operator experience in the training program [15] and proven to be an independent predictor of adverse outcome after neuro-intervention [24]. However, most of these neurovascular simulators offer limited coverage of vascular anatomy. A complete vascular simulator with both extracranial and intracranial vascular components is still lacking. 

To diminish the radiation dose under training procedures, an ambulatory real-time infrared imaging platform (IRIP) with rotatory C-arm has been developed in our lab and applied for imaging guidance instead of the traditional fluoroscope in the angio-suite. With this infrared imaging platform, the heavy protective apron is no longer necessary for the trainees. The simulation procedure could be held in a meeting room and no longer confined to the angio-room. Herein, we share our experience of the full-sized patient-specific 3D extracranial VS and the IRIP for the training of residents to improve their performance and safety in neuroendovascular procedures.

## 2. Materials and Methods

This retrospective study was approved by the Institutional Review Board of our hospital (IRB Serial No. 10905-011). The requirement to obtain informed consent was waived due to the study’s retrospective nature.

### 2.1. Vascular Simulators

We retrospectively reviewed the picture archiving and communication system (PACS) of our institution for subjects who received thoracoabdominal computed tomography (CT) angiography for the preoperative evaluation of transcatheter aortic valve implantation (TAVI) from January 2017 to June 2018. As a result, one case with type I aortic arch, one case with bovine variant, and another two cases with type III aortic arch were enrolled for the 3D vascular segmentation to fabricate the patient-specific 3D extracranial vascular simulators for the resident training of neuroangiography techniques (Figure 2). The type I aortic arch model was used for the introduction of this simulation system to the trainees. The bovine variant model was used for the manipulation of the Simmon 2 (SIM2) catheter, which is the choice of catheter for type III aortic arch. The type III aortic arch model with tortuous abdominal aorta was used as the final test of trainee’s performance to manage difficult aortic arch. These 3D-print VSs had been used in the neuro-interventional training course for the residents of our hospital.

In this project, a total of 4 full-sized carotid-aortic-iliac models were printed by an X-CUBE Stereolithography (SLA) 3D printer using SLA material. Below is the summary of the workflow of 3D image segmentation and build-up of the 3D-print model through use of 3D calculation software, Soft Mimics17.0 (Materialise, Leuven, Belgium). At first, the thoracoabdominal CT scans for pre-TAVI evaluation were performed on a dual-source CT scanner (Definition Flash; Siemens Healthcare, Forchheim, Germany). A high-pitch spiral CT angiography from the aortic arch to the femoral arteries was carried out with the following parameters: tube voltage 80 kVp; reference tube-current-time product 400 mAs; rotation time 0.28 s; slice collimation 128  mm ×  0.6 mm; and pitch value 3.0. All patients received one bolus of nonionic iodinated contrast medium (Optiray, Mallinckrodt Medical Inc., St. Louis, MO, USA) with a total volume of 53 mL and a flow rate of 4 mL/s. Then, the raw images were exported in standard digital imaging and communication in medicine (DICOM) format to the 3D calculation software, Soft Mimics17.0 (Materialise, Belgium). The 3D extracranial vascular model was segmented below the origin of the superficial femoral artery to the supra-aortic regions with preserved common carotid artery stumps and a minimal length of 15 cm. The segmented data were transformed into a stereolithography (STL) file and exported from Soft Mimics to 3-Matics (Materialise, Belgium). The unnecessary part of the model was trimmed with the relevant vascular channels kept available in the residual 3D model for the need of catheter navigation. Finally, the solid vascular model was transformed to the hollow channel by postprocessing steps in the following order: (1) fix the surface with Fix Wizard automatically; (2) smooth the surface with the factor of 0.7; (3) hollow the vascular lumen with imaginary thickness 1 mm outside to obtain a vascular wall image; (4) fabricate the 3D hollow vascular model with the transparent SLA material, an acrylate-based polymer resin (MSLA modeling clear, ApplyLabWork, Torrance, CA, USA) with a flexure modulus of 1800–2000 and a hardness Shore D of 85–88; and (5) verify the fidelity of the whole vascular simulator under fluoroscopy by catheter navigation. Filling the whole VS with normal saline is required to enhance the effect of hydrophilic coating on the diagnostic catheters/wires and to lower the friction during catheter manipulation.

### 2.2. Infrared Imaging Platform 

To provide a training platform without radiation exposure, one infrared imaging system was constructed with a rotatory C-arm and a 60 cm × 115 cm × 150 cm platform (Figure 3a). The diameter of the C-arm was 70 cm. There was one infrared source of 850 nm at one end of the C-arm and one infrared image capture device at another end. The image capture device was composed of one infrared camera (Grasshopper U3, Teledyne FLIR, Wilsonville, OR, USA), one 850 nm filter, and one C-mount lens (M0814-MP2, Computar, Tokyo, Japan) with a fixed focal length of 8 mm and an aperture range from F1.4 to F16. The camera consisted of complementary metal-oxide semiconductor (CMOS) sensors with a sensor format of 1/1.2”, a 163 frame rate, and a resolution of 1920 × 1200. The whole image capture device was 40 mm × 45 mm × 120 mm in dimensions. Between two ends of the C-arm, one 45 cm × 30 cm light guide plate integrated at one half of the platform was placed at the center of the infrared light path. The infrared light passing through the light guide plate, vascular simulator, and catheter/wire were captured by the image capture device. Using the application programming interface (SpinView, Teledyne FLIR, Wilsonville, OR, USA), the infrared camera took images in a continuous acquisition mode with an exposure time of 4.5 ms and an acquisition frame rate of 163. The images were displayed on a 24-inch monitor. Figure 3b shows the screen capture of the infrared image during manipulation of a SIM2 catheter in a vascular simulator with a type I aortic arch.

### 2.3. The Two Training Courses in the Past Four Years

In the past 4 years, two training courses using the vascular simulators were held in our department. The first course took place in March 2018, with the traditional angiography machine (AM) being used as the imaging platform. A total of 10 residents participated in the first course, including three first-year residents (R1s) with no experience in angiography, three second-year residents (R2s) with experience in abdominal angiography, and four third-year residents (R3s) with experience in abdominal and cerebral angiography (Figure 4). In March 2022, the second course was implemented with a traditional angiography machine and an infrared imaging system as the imaging platforms. A total of 9 participants joined the second course, including three R1s, three R2s, and three R3s. 

Figure 5 provides the summary of the training program and the final test using the vascular simulators.

Basic introduction of this training program was given to all participants, including (1) the three VSs with different aortic arch types; (2) the manipulation techniques of 4 French (Fr) JB1, JB2, and SIM2 catheters; and (3) the two videos of JB2 catheter manipulation in one type I aortic arch simulator (Appendix A) and SIM2 catheter manipulation coupled with a long 6 Fr curved angiosheath to overcome the tortuosity of the abdominal aorta in one type III aortic arch (Appendix A).After the basic introduction, all participants were separately invited to experience and practice the catheter manipulation with different aortic arch simulators under fluoroscopy (first course) or the infrared system (second course). One 5 Fr sheath had been already introduced into the right common femoral artery of each simulator. With a Siemens Artis zee biplane angiography system (Siemens, Munich, Germany), low-dose radiation fluoroscopy settings were applied for the training program with a reduced frame rate (5 fps) and a lower x-ray dose per frame (29–32 nGy/frame). Each resident was asked to complete the navigation of one 4F r JB1 catheter to right the CCA/left CCA on a type I aortic arch simulator and advancement of one 4 Fr SIM2 catheter to the right CCA/left CCA on a bovine arch simulator step by step.For the type III aortic arch simulator with tortuous abdominal aorta, each trainee was asked to experience the difficulty of the JB2 catheter navigation to right the CCA and SIM 2 catheter reconstitution in the aortic arch of this specific simulator. Then, a 60 cm 6 Fr curved angiosheath was introduced to the thoracoabdominal aorta junction to overcome the redundant movement of the SIM catheter during reconstitution. All trainees were allowed to complete a practice navigation of the right CCA and left CCA with a SIM 2 catheter at this point. Finally, the time duration of the navigation of the right CCA and left CCA with the SIM 2 catheter was defined as the total manipulation time (TMT) and recorded for all trainees. In the first training course, all four third-year residents were allowed to have only one trial, and the other second-year and first-year residents were allowed to have two trials. In the second course, all residents were allowed to have two trials.

### 2.4. Statistical Analysis

The total manipulation times are expressed as mean or median with range for all data. The effect of the two training courses was estimated with the TMT of each participant and the change of TMT between the first trial and second trial. Due to the small sample size of this study, the training effect was compared in the following ways with different statistical tests: (1) Mann–Whitney *U* test for the intergroup analysis of TMT and TMT change to compare AM vs. IRIP and junior participants (R1) vs. senior participants (R2 and R3); (2) Wilcoxon signed-rank test for the analysis of the TMT between the first and second trial of each participant. Values of *p* value < 0.05 were considered to indicate statistical significance.

## 3. Results

The total manipulation time of all participants in the first and second training courses is listed in Table 1.

In the first training course using a traditional AM, the average TMT of the first trial was 13.1 min (range 8.5~18.0 min; median 13 min) for 3 R1s, 3 R2s, and 4 R3s. Among the 3R1s and 3 R2s attending the second trial, the average TMT of second trial was 3.4 min (range 2.5~4.8 min; median 3.1 min), which was less than that of the first trial. In the second training course using the IRIP, the average TMT of the first trial was 6.7 min (range 3.6 to 14.0 min; median 5.7 min) for another 3 R1s, 3 R2s, and 3 R3s. For all 9 participants, the average TMT of the second trial was 1.9 min (range −0.8 to 6.6 min; median 1.4 min), which was less than that of the first trial. The TMT of the second trial (range 2.2~14.4 min; median 5.9 min) was significantly shorter than that of the first trial (range 3.6~18 min; median 8.7 min, regardless of whether the AM or the IRIP was used (Wilcoxon signed-rank rest, *p* = 0.001; Figure 6a). For the first trial, the TMT was significantly shorter in the second training course using the infrared system than that using the angiography machine (Mann–Whitney *U* test, *p* = 0.003; Figure 6b). The mean TMT improvement for the six R1s was 3.7 min (range −0.8–6.6 min; median 4.1 min), which was significantly better than that for the 6 R2s and 3 R3s of 1.7 min (range 0.4–2.6 min; median 1.4 min; *p* = 0.049; Figure 6c). For the six R1s, the mean TMT of the second trial was reduced to 9.9 min (range 4.9–14.4 min; median 10.3 min). For the senior residents (R2s and R3s), the mean TMT of the first trial and second trial was 8.4 min (range 3.6–14.1 min; median 8.5 min) and 6.4 min (range 2.2–11.5 min; median 5.8 min), respectively. Based on the TMT, the performance of the junior residents (R1s) at the second trial showed no significant differences compared to that of the senior residents (R2s and R3s) in the first and second trials (*p* = 0.453 and 0.067 respectively; Mann–Whitney *U* test).

According to the responses of these trainees, the major advantages of this simulation system are that (1) it can provide a composite practice of the angiography machine and catheter/wire manipulation, (2) it can provide interactive tactile feedback secondary to the friction and resistance between the catheter/wire and arterial wall, (3) it can facilitate repeated experience of the SIM 2 catheter manipulation in a difficult aortic arch, and (4) it can increase the confidence of trainees when dealing with a difficult aortic arch. As for the infrared system, all three R1s and three R2s attending the second training course preferred the infrared system as the training platform. On the other hand, two out of three R3s preferred the angiography machine due to the sharper fluoroscopic imaging of the wires and catheters. 

## 4. Discussion

In this study, we fabricated four extracranial vascular simulators with different aortic arch types and an infrared imaging system as a neuro-interventional training platform. Owing to the didactic lectures and step-by-step practice on this training platform, junior residents with little neuro-angiographic experience could reach a similar performance to the senior residents. The patient-specific VS could serve as an elementary platform in a neuro-interventional training program for novices with little and even no angiographic experience.

Despite the rapid advancement of noninvasive CT and magnetic resonance angiography in the past decades, traditional diagnostic neuroangiography remains the gold standard for various neurovascular diseases and the cornerstone of neurovascular intervention treatments. Being an invasive procedure, however, diagnostic cerebral angiography carries a low but significant risk (about 1%) of ischemic events with 0.1% permanent morbidity [24]. The incidence of neurologic events after the carotid angiography is even higher in the setting of cerebrovascular diseases [24], aging patients (>80 years old) [7], difficult arch anatomies (type III and bovine variants) [25,26,27,28], and less-experienced operators [5,29]. The impact of difficult arch anatomies on carotid artery stenting (CAS) has become more prominent [7,25,26,27,28,29]. Type III aortic arch has been proven to be an independent risk factor for complications and difficulties of CAS [26,27]. In the study of Burzotta, et al., the combined prevalence of type III and bovine variants was as high as 30% in the CAS population [28]. Therefore, type III aortic arch and the bovine configuration were chosen as the training vascular simulators.

Total fluoroscopic time [5,6] and catheter manipulation time [28,29] have also been proposed as objective markers for the operator’s experience and another independent risk factor for the unfavorable outcome of CAS procedures. In our study, the total manipulation time on a VS of type III arch, calculated from the long angiosheath setup to the finish of left CCA cannulation using a SIM2 catheter, was used for the measurement of the trainee’s skills. The result of our study showed the shortening of TMT at the second trial could denote improvement of the trainee’s skills and experience. The total fluoroscopic time of the final test was not a target measurement of our study due to several concurrent tasks on the type III arch simulator, including experience of difficulties of carotid artery navigation using JB2 catheter, SIM2 catheter reconstitution in the type III arch, and setup of a long sheath to overcome the tortuosity of thoracoabdominal aorta before the final test. In Miranpuri’s study [15], use of a silicone vascular model with generalized arch configuration for basic neuroangiography simulation significantly decreased both overall procedure time and total fluoro time of 10 residents after repeated practice. Apart from the skill of catheter manipulation and vessel navigation, the total fluoro time is also a reflection of the control of the angiography machine. In our training platform, the vascular model with type I arch configuration plays a similar role for the composite experience of catheter manipulation and angiography machine control. Therefore, the vascular models with bovine arch and type III arch configurations were used in the advanced training course for complex arch patterns.

Even with the same practice plan used before the final test, the TMT of the first trial was significantly shorter in the second training course using the IRIP than in the first course using the AM. This could be due to the following reasons. First, the infrared system could provide another platform to release the crowding of the whole training process. With the ambulatory design of IRIP, the training program could be held in a conference room. The participants could have more time for practice before the final test, especially the junior trainees with little experience in neuroangiography. Second, the infrared system could provide a platform that does not require radiation protection. The trainees could practice comfortably without wearing a heavy apron. Compared with the first training course, the junior trainees could have more time to familiarize themselves with catheter manipulation. 

Since the publication of pivotal trials in 2015 [30,31,32,33,34], endovascular thrombectomy (EVT) has been an important treatment of choice for acute ischemic stroke with level 1A evidence [35]. Even though there was a large jump in the annual number of EVT procedures in 2015, fewer than half of acute stroke patients who are potentially EVT eligible can receive this procedure [36,37,38]. According to the DAWN [39] and DEFUSE-3 [40] trials, the case burden of EVT to the stroke systems of care and the number gap between “treat” and “nontreat” of EVT eligible patients will continue to increase while the treatment window is extended from 8 h to 16~24. The lack of qualified neuro-interventionalists is a barrier for the real-world practice [37,41]. For a physician to become a competent and safe examiner for the carotid and intracranial vasculature, at least 200 diagnostic cerebral angiograms are necessary as the training requirement [5,42]. Thereafter, another 200 selective vascular catheterizations are the prerequisite to becoming qualified in performing the endovascular treatment for acute ischemic stroke [42]. A 5-year interval is usually needed to complete the specific training program. For the less common but challenging aortic arches with type III or bovine configurations, a large amount of cerebral angiogram practice is compulsory for the training program. With the assistance of extracranial vascular models with various arch configurations as a training platform, the number of cerebral angiograms to fulfill the training requirement might be reduced without a compromise of technical skills. 

In a transparent vascular simulator, catheter manipulation and vessel cannulation could be performed under direct visualization or reverted to a videoscopic mode that can be beneficial for endovascular training [43]. Compared to direct visualization or the videoscopic mode, however, the whole simulation experience under fluoroscopic guidance is more similar to a routine endovascular procedure. If radiation exposure is not necessary for the trainees and proctors, an infrared imaging system would be a potential future direction in designing a pragmatic endovascular training platform. Additionally, the infrared imaging system could be a practical demonstration platform for endovascular devices manufacturers.

## 5. Conclusions

Compared to the traditional training program for diagnostic cerebral angiography, patient-specific vascular simulators with different aortic arch types could rapidly enrich the experience of the young trainees. Coupling the simulator with an ambulatory infrared imaging system could provide a realistic and radiation-free platform for neurovascular training, especially for those junior residents with little experience in neuroangiography. The training program implemented with patient-specific vascular simulators and the infrared imaging platform has the potential to maintain effective training for neuro-interventional procedures in a shorter period without compromising technical skills. 

## Figures and Tables

**Figure 1 healthcare-10-02277-f001:**
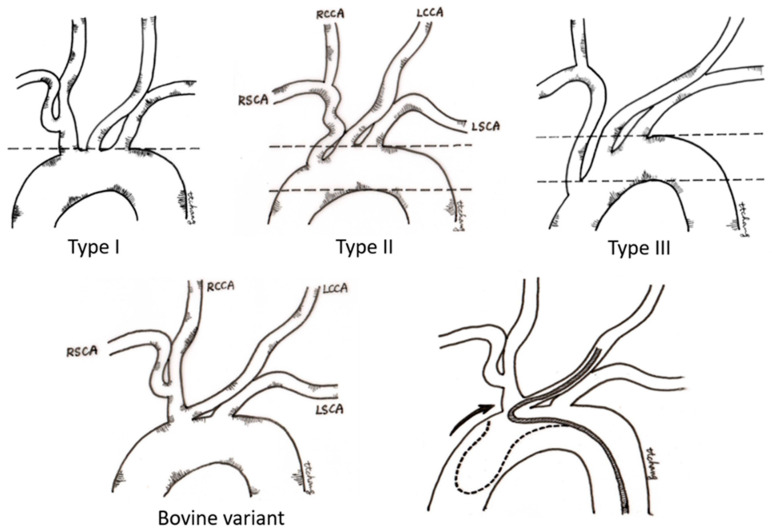
Classification of the aortic arch. In the type I aortic arch, the openings of all three major vessels are on the horizontal line of the upper curvature of the aortic arch. In the type II aortic arch, the innominate artery originates between the upper and lower curvature of the aortic arch. In the type III aortic arch, the origin of the innominate artery is below the horizontal line of the lower curvature of the aortic arch. For the type II and type III arch, access to the innominate trunk and left common carotid artery becomes increasingly difficult because a catheter approaching from the descending aorta tends to prolapse into the ascending aorta. In the bovine variant of aortic arch, the left common carotid artery originates from the innominate artery, and catheter navigation to the left common carotid artery usually requires special techniques of SIM2 catheter manipulation. (LCCA—left carotid artery; LSCA—left subclavian artery; RCCA—right common carotid artery; RSCA—right subclavian artery).

**Figure 2 healthcare-10-02277-f002:**
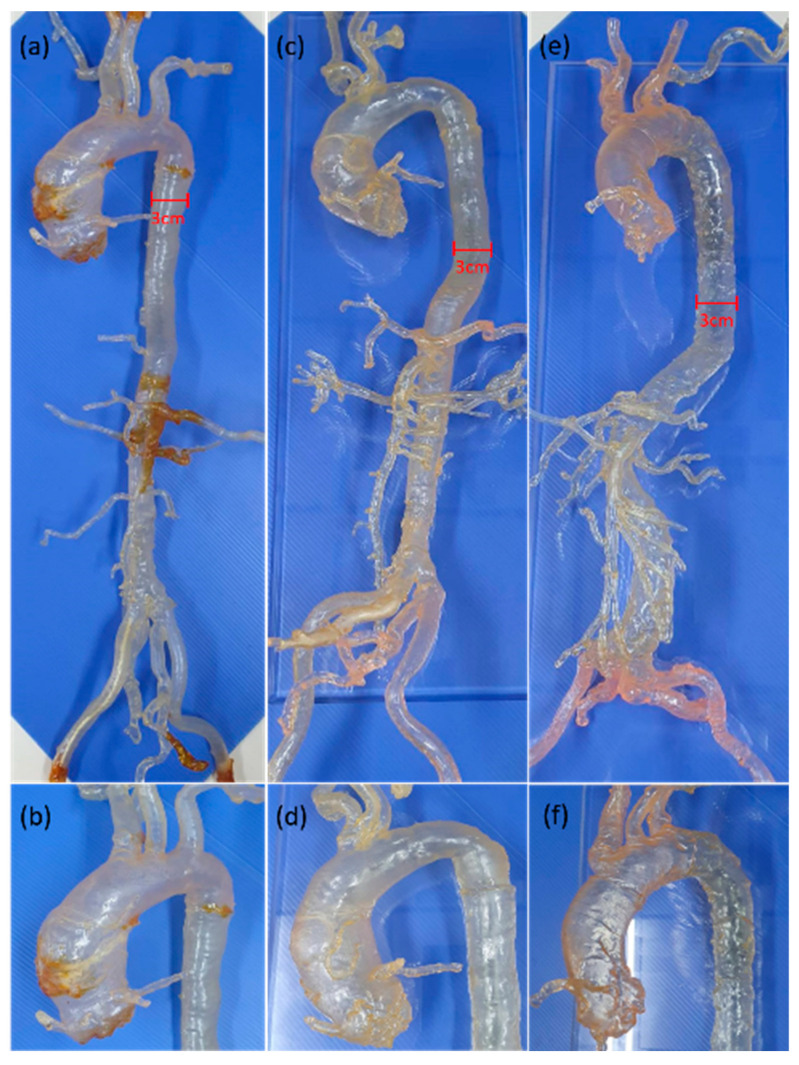
Whole picture of full-sized carotid-aortic-iliac vascular simulators and a magnified view of the aortic arch. (**a**,**b**) Type I arch; (**c**,**d**) bovine variant; (**e**,**f**) type III arch.

**Figure 3 healthcare-10-02277-f003:**
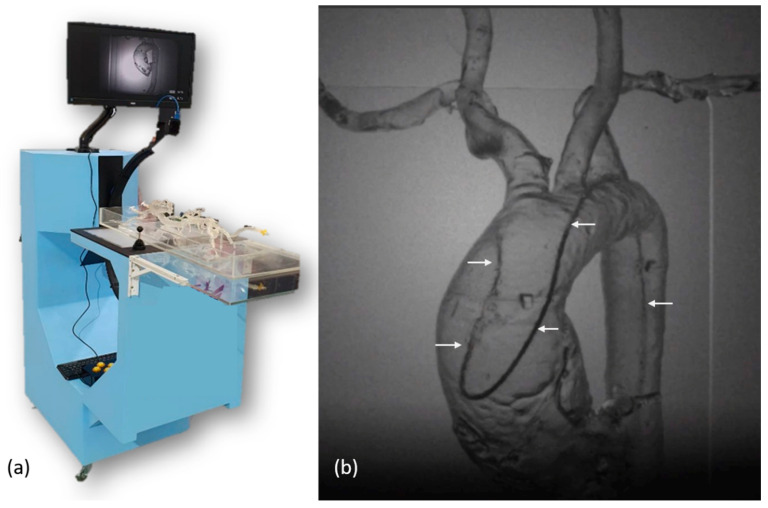
(**a**) Photo of the infrared imaging platform; (**b**) The infrared image of a SIM2 catheter (arrows) in a vascular simulator with a type I aortic arch.

**Figure 4 healthcare-10-02277-f004:**
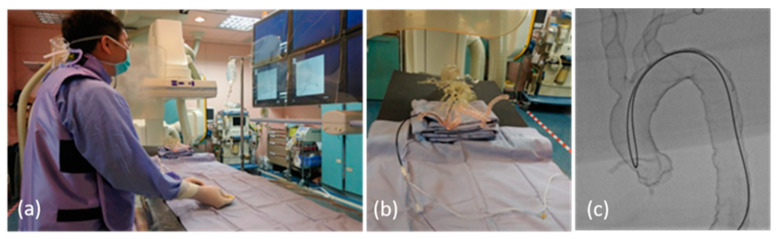
(**a**) Photo of the training program using the vascular simulator in the angio-suite; (**b**) photo of the vascular simulator setting in the angio-suite; (**c**) the fluoroscopic image of the SIM2 catheter manipulation in a vascular simulator with type III aortic arch.

**Figure 5 healthcare-10-02277-f005:**
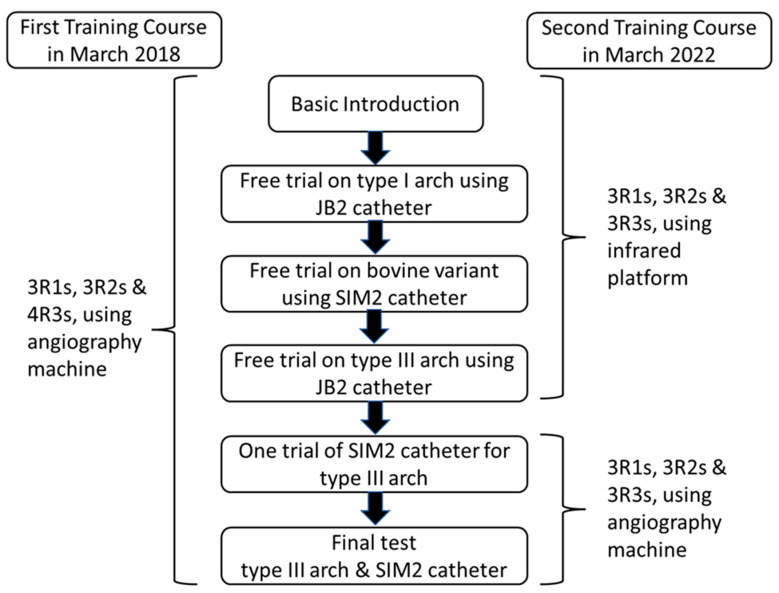
Process of the first and second training courses.

**Figure 6 healthcare-10-02277-f006:**
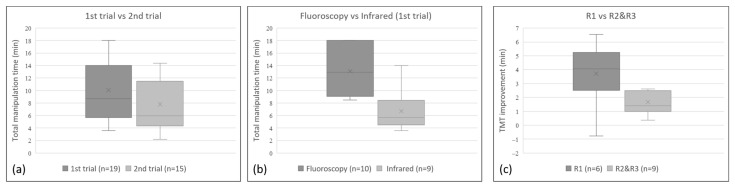
Statistically significant differences (*p* < 0.05) can be observed in the box plot of (**a**) the TMT of the first trial vs. that of second trial for all participants—showing that the TMT of the second trial was significantly shorter than that of first trial—(**b**) the TMT of the first trial using fluoroscopy versus the infrared system—showing a significantly shorter TMT using the infrared system than that using the fluoroscopy—and (**c**) the TMT improvement of junior participants (R1) versus senior participants (R2 and R3)—showing a significantly larger improvement in the TMT of the junior participants (R1) than that of senior participants (R2 and R3).

**Table 1 healthcare-10-02277-t001:** Total manipulation time of all participants in the two training courses.

	First Trial (min)	Second Trial (min)	Improvement (min)	Ratio
Fluoro R1-A	18.0	13.7	4.3	23.9%
Fluoro R1-B	18.0	13.2	4.8	26.7%
Fluoro R1-C	18.0	14.4	3.6	20.0%
Fluoro R2-A	12.7	10.2	2.5	19.7%
Fluoro R2-B	14.1	11.5	2.6	18.4%
Fluoro R2-C	13.2	10.7	2.5	18.9%
Fluoro R3-A	8.5			
Fluoro R3-B	8.6			
Fluoro R3-C	10.5			
Fluoro R3-D	9.2			
IR R1-A	5.2	5.9	−0.7	−13.5%
IR R1-B	14.0	7.5	6.5	46.4%
IR R1-C	8.7	4.9	3.8	43.7%
IR R2-A	4.9	3.5	1.4	28.6%
IR R2-B	8.2	5.9	2.3	28.0%
IR R2-C	5.7	4.3	1.4	24.6%
IR R3-A	6.2	5.8	0.4	6.5%
IR R3-B	3.6	2.2	1.4	38.9%
IR R3-C	4.2	3.5	0.7	16.7%

IR—infrared system; Fluoro—fluoroscopy.

## Data Availability

The complete data are available from the corresponding author on reasonable request.

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
