# Peer review of "Patient-Specific 3D-Print Extracranial Vascular Simulators and Infrared Imaging Platform for Diagnostic Cerebral Angiography Training"

_healthcare, 2022, doi:10.3390/healthcare10112277_

Round 1

Reviewer 1 Report

This study aimed at their experience of using 3D-printed extracranial vascular simulators and the infrared imaging platform in two training courses for diagnostic cerebral angiography. Patient-specific VS with radiation-free IRIP could be a useful training platform for junior residents with little experience in neuroangiography.

With the advancement of 3D-print techniques, the medical simulation model is becoming an important modality in the neurosurgical training nowadays, such as neurovascular surgery and aneurysm microsurgery.

There had fabricated four extracranial vascular simulators with different aortic arch types and an infrared imaging system as a neuro-interventional training platform. Combined with the didactic lectures and step-by-step practice on this training platform, junior residents with little neuro-angiographic experience could reach the similar performance of the senior residents. This study analysis the training effect with Mann-Whitney U test for intergroup analysis from aspects of TMT change in AM vs. IRP and R1 vs. R2&R3, as well as Wilicoxon Signed-Rank Test for TMT between 1st and 2nd of 19 participants, the results could support the opinion that 3D-print extracranial vascular simulators with infrared imaging system could serve as a neuro-interventional training platform. Insufficient number of participants as well as lack of 3rd trial are drawbacks, it is proposed to compare within group differences of R1R2 and R3 respectively, which can allow conclusions more powerfully before publishing the study.

There is a minor mistakes:In the 65 linemissing a mark.

Reviewer 2 Report

Major comments:

Please explain in the introduction why the time to perform these tasks are relevant/important, and ideally quantifies the effects of time.

On line 227, the authors claim that the times for junior residents is similar to senior, but when looking at fig 6 c, it looks like the R2 and R3 perform better, not similar. Please explain/rephrase.

The conclusions are very short. The authors can perhaps add also short summary and why the IRIP is particularly suitable for junior residents. What is more concerning is that this study might be so limited that it is not really possible to draw any conclusions.

Minor comments:

In the abstract it is unclear if IRIP or AM or both are used and what is the difference between the two courses. It is much clearer in fig 5, but consider rephrasing in the abstract.

Are IRP and IRIP the same?

The keywords are too broad and may be misleading: Suggested changes are e.g. infrared imaging instead of infrared, simulator instead of simulation, add e.g. neuro-interventionalist

Line 37: treatment mainstream -> mainstream treatment?

Please introduce angiosuite.

Figure 1 is good. For even better readability, can one more schematic be added illustrating the catheter navigation discussed in the caption? Consider adding labels of the different arteries.

Line 76: Can you add a reference for the IRIP with rotary C-arm? Or is that technology not published before? Or clarify that the rotary C-arm is part of this publication.

Is there no conflict of interest because the authors are involved in the training course?

What are the material properties of the 3D simulator compared to actual situation? E.g. softness, durability, friction

Scale bars are missing in figure 2.

Consider including a schematic in fig 3 illustrating the content in the paragraph starting line 126. Alternatively, add explanatory text and arrows in fig 3a.

Line 137 yours -> years?

Add a topic sentence on line 153. The pullet point list comes very sudden.

Was the second course better somehow because of experience from the first course that would contribute to the difference in TMT?

Consider changing the order in table 1 to align with text.

The list of abbreviations is not complete. In the text you mention PACS, CT, X-CUBE, CTA, CM, DICOM, STL, IRP, MR, CAS, CMT, CCA, JB2, EVT and possibly more.

Please use the full word and abbreviation in brackets only the first time it is mentioned in the text. For instance, TMT has this several times.

Why is the corresponding author email address privat (gmail)?

Reviewer 3 Report

Tortuous aortic arch is always challenging for beginners of neuro-interven-tionalists. The authors share their experience of using 3D-printed extracranial vascular simulators (VS) and the infrared imaging platform (IRIP) in two training courses for diagnostic cerebral angiography in the past 4 years. It is a very interesting interdisciplinary work including AM, medicine and materials. Some questions need to be answered by the authors.

1.      The scale bars should be listed in the Fig. 2.

2.      The information about SLA materials should be fully stated. What is the name and composition or source of the 3D printing materials? And what is the modulus of the printed materials?

3.      The English writing in some parts should be improved.

4.      The statement of Conclusion is too simple and vague. Please reconsider it.
